# MerMAIDs: a family of metagenomically discovered marine anion-conducting and intensely desensitizing channelrhodopsins

Johannes Oppermann [1], Paul Fischer[1], Arita Silapetere[1], Bernhard Liepe[1], Silvia Rodriguez-Rozada [2], José Flores-Uribe [3,6], Enrico Schiewer[1], Anke Keidel[4], Johannes Vierock[1], Joel Kaufmann[5], Matthias Broser[1], Meike Luck[1], Franz Bartl[5], Peter Hildebrandt[4], J. Simon Wiegert [2], Oded Béjà[3], Peter Hegemann [1] & Jonas Wietek [1,7]

Channelrhodopsins (ChRs) are algal light-gated ion channels widely used as optogenetic tools for manipulating neuronal activity. ChRs desensitize under continuous bright-light illumination, resulting in a significant decline of photocurrents. Here we describe a metagenomically identified family of phylogenetically distinct anion-conducting ChRs (designated MerMAIDs). MerMAIDs almost completely desensitize during continuous illumination due to accumulation of a late non-conducting photointermediate that disrupts the ion permeation pathway. MerMAID desensitization can be fully explained by a single photocycle in which a long-lived desensitized state follows the short-lived conducting state. A conserved cysteine is the critical factor in desensitization, as its mutation results in recovery of large stationary photocurrents. The rapid desensitization of MerMAIDs enables their use as optogenetic silencers for transient suppression of individual action potentials without affecting subsequent spiking during continuous illumination. Our results could facilitate the development of optogenetic tools from metagenomic databases and enhance general understanding of ChR function.

---

[1] Institute for Biology, Experimental Biophysics, Humboldt-Universität zu Berlin, Invalidenstraße 42, 10115 Berlin, Germany. [2] Research Group Synaptic Wiring and Information Processing, Center for Molecular Neurobiology Hamburg, Falkenried 94, 20251 Hamburg, Germany. [3] Technion—Israel Institute of Technology, 32000 Haifa, Israel. [4] Institute for Chemistry, Physical Chemistry/Biophysical Chemistry, Technische Universität Berlin, Straße des 17. Juni 135, 10623 Berlin, Germany. [5] Institute for Biology, Biophysical Chemistry, Humboldt-Universität zu Berlin, Invalidenstraße 42, 10115 Berlin, Germany. [6] Present address: Department of Plant Microbe Interactions, Max Planck Institute for Plant Breeding Research, Cologne 50829, Germany. [7] Present address: Department of Neurobiology, Weizmann Institute of Science, 7610001 Rehovot, Israel. Correspondence and requests for materials should be addressed to P.H. (email: hegemann@rz.hu-berlin.de) or to J.W. (email: jonaswietek@gmail.com)

Channelrhodopsins (ChRs) are members of the microbial rhodopsin family that directly translate absorbed light into ion fluxes along electrochemical gradients across cellular membranes by opening a conductive pore[1–3]. ChRs are composed of seven transmembrane helices and an embedded retinal cofactor linked to a conserved lysine in helix 7 via a Schiff base (retinal Schiff base, RSB). Upon photon absorption, the RSB isomerizes from all−trans to 13-cis, which induces structural changes, collectively described as spectroscopically distinguishable intermediates in a photocycle[4].

In response to extended light pulses, the photocurrents of most known ChRs decline from an initial peak current to a lower, stationary level, a phenomenon known as desensitization (also termed inactivation)[2,4–6]. The degree and kinetics of desensitization differ among ChRs and depend on pH, membrane voltage as well as light intensity and color, with typically ≤70% amplitude reduction[1,2,7]. Photocurrent decrease via desensitization has been explained by accumulation of late non-conducting photocycle intermediates and by an alternative photocycle exhibiting low cation conductance[7–10].

During the past fourteen years, cation-conducting ChRs (CCRs) were widely employed to depolarize genetically targeted neurons or neuronal networks using light to trigger action-potential firing[11–15]. Originally, light-driven microbial ion pumps were utilized to suppress neuronal activity by hyperpolarization[16,17]. Since ion pumps always transport one ion per absorbed photon, efficient neuronal silencing required high ion pump expression levels and continuous, intense illumination. This disadvantage was overcome by converting CCRs into anion-conducting ChRs (ACRs)[18–21]. Such engineered ACRs (eACRs)[22] and later−discovered natural ACRs (nACRs)[23–26] silence neuronal activity by light-induced shunting-inhibition, similar to endogenous GABA- or glycine-activated chloride channels[22,27–30].

Here, we report a family of phylogenetically distinct ChRs metagenomically identified from marine microorganisms. These ChRs conduct anions but exhibit unique desensitization in continuous light and are therefore designated MerMAIDs (Metagenomically discovered, Marine, Anion-conducting and Intensely Desensitizing ChRs). Seven MerMAIDs are characterized biophysically via electrophysiological recordings, and we elucidate the molecular mechanism of the first accessible MerMAID using spectroscopic analyses and molecular dynamic (MD) calculations. We also explore the optogenetic inhibitory potential in neurons.

## Results

**A channelrhodopsin family with distinct desensitization.** Seven putative ChRs constituting a not yet described and distinct phylogenetic branch in the ChR superfamily were identified in contigs assembled from the *Tara* Oceans metagenomes (MerMAIDs in Fig. 1a). However, the shortness of the assemblies (<10 kb) precluded taxonomic classification of the contigs. These MerMAIDs appeared to be globally distributed in the oceans, most abundant at stations near the equatorial Pacific and South Atlantic Oceans (Fig. 1b). The MerMAIDs were primarily constrained to the photic zone (depth, 0–200 m), as previously reported for other rhodopsins[31] (Supplementary Fig. 1).

Phylogenetically, the MerMAIDs appear more closely related to chlorophyte CCRs than cryptophyte ACRs (Fig. 1a). Sequence comparisons, however, indicated that MerMAIDs might be anion-conducting due to the lack of typical glutamate residues found in chlorophyte CCRs (Supplementary Fig. 2). As already shown, replacement of pore-lining glutamates with positively charged or neutral amino acids can mediate anion selectivity in originally cation-conducting chlorophyte CCRs[18–22]. Nevertheless, cryptophyte CCRs were shown to conduct cations although

lacking typical glutamate motives by operating with an alternative mode more related to light-driven rhodopsin ion pumps[32–34]. To examine the MerMAIDs function and ion selectivity, we expressed them in human embryonic kidney (HEK) cells and performed whole-cell voltage-clamp experiments at 1-day post-transfection.

When excited with 500-nm light, MerMAID-expressing cells exhibited large photocurrents but in contrast to all previously analyzed ChRs (Fig. 1c), MerMAIDs reveal almost complete desensitization with continuous, bright light exposure (Fig. 1c, d and Supplementary Fig. 3a, b). Maximum peak photocurrent amplitudes reached up to 2 nA (MerMAID6, Fig. 1e and Supplementary Fig. 3c), but the current did not saturate even at 3.73 mW/mm2 (Supplementary Fig. 3d). Transient photocurrent action spectra were recorded to determine the wavelength sensitivity of the MerMAIDs. All variants tested exhibited typical rhodopsin spectra, with maximal sensitivity close to 500 nm (Fig. 1f, g and Supplementary Fig. 3e), as expected for marine organisms, given that blue light penetration is strongest within the photic zone in seawater (Fig. 1f).

**MerMAIDs selectively conduct anions.** Next, we tested the ion selectivity of the MerMAIDs. Therefore, photocurrents at different membrane potentials were recorded to deduce the reversal potential ($E_{rev}$; the potential where the net ion flux is zero) in different ionic conditions (Fig. 2a, b). Because we suspected anion selectivity, we depleted the extracellular $Cl^-$ from 150 mM to 10 mM while maintaining the intracellular $Cl^-$ at 120 mM (Fig. 2a). This increased the inward current and induced a positive shift of the reversal potential ($\Delta E_{rev}$, Fig. 2a, b), consistent with a $Cl^-$ outward flux. A similar shift close to the theoretical $Cl^-$-Nernst potential was obtained for all MerMAIDs (Fig. 2c and Supplementary Fig. 4a), as well as for the small stationary photocurrents of MerMAID1 (Fig. 2a, c and Supplementary Fig. 4a). These data justified the classification of MerMAIDs as ACRs.

To evaluate the conductance of other anions, we performed ion substitution experiments using MerMAID1 as a model. Replacement of $Cl^-$ with $Br^-$ or $NO_3^-$ resulted in negative reversal potential shifts (Fig. 2d and Supplementary Fig. 4b), thus revealing nonselective anion conductivity with a relative permeability sequence that follows $Cl^- < Br^- < NO_3^-$, as previously reported for other ACRs[21,23]. In contrast, substitution of $Na^+$ with $NMDG^+$, $Ca^{2+}$, or $Mg^{2+}$ had only a slight effect on reversal potentials (Fig. 2d and Supplementary Fig. 4b), thereby excluding a substantial contribution by cations as charge carriers.

**Photocurrent properties are unaffected by pH-changes.** Rhodopsin function often involves de- and reprotonation of internal amino acids, and pH changes can significantly affect photocurrent amplitude and kinetics[4,9,18,19]. We therefore investigated the effect of extra- and intracellular pH ($pH_e$ and $pH_i$) changes on MerMAID1. Variation of $pH_e$ between 6.0 and 8.0 slightly altered the photocurrent amplitude (Supplementary Fig. 4c) but not the reversal potential (Fig. 2d and Supplementary Fig. 4d), thus excluding proton transport. Neither $pH_i$ nor $pH_e$ affected the desensitization time constant, $\tau_{des}$ (Fig. 2e, f and Supplementary Fig. 4e–g). However, $\tau_{des}$ exhibits a moderate voltage dependence (Fig. 2f, g and Supplementary Fig. 4e, f). For MerMAID1,3–5, desensitization slowed down with increasing membrane potential whereas $\tau_{des}$ decreased for MerMAIDs 2, 6, and 7 (Fig. 2g). These groups correlated well with the two phylogenetic branches within the MerMAID family (Fig. 1a), although the underlying molecular determinants of this difference remain unknown.

To assess the photocycle turnover time (recovery kinetic time constant, $\tau_{rec}$), we performed double-pulse measurements at −60

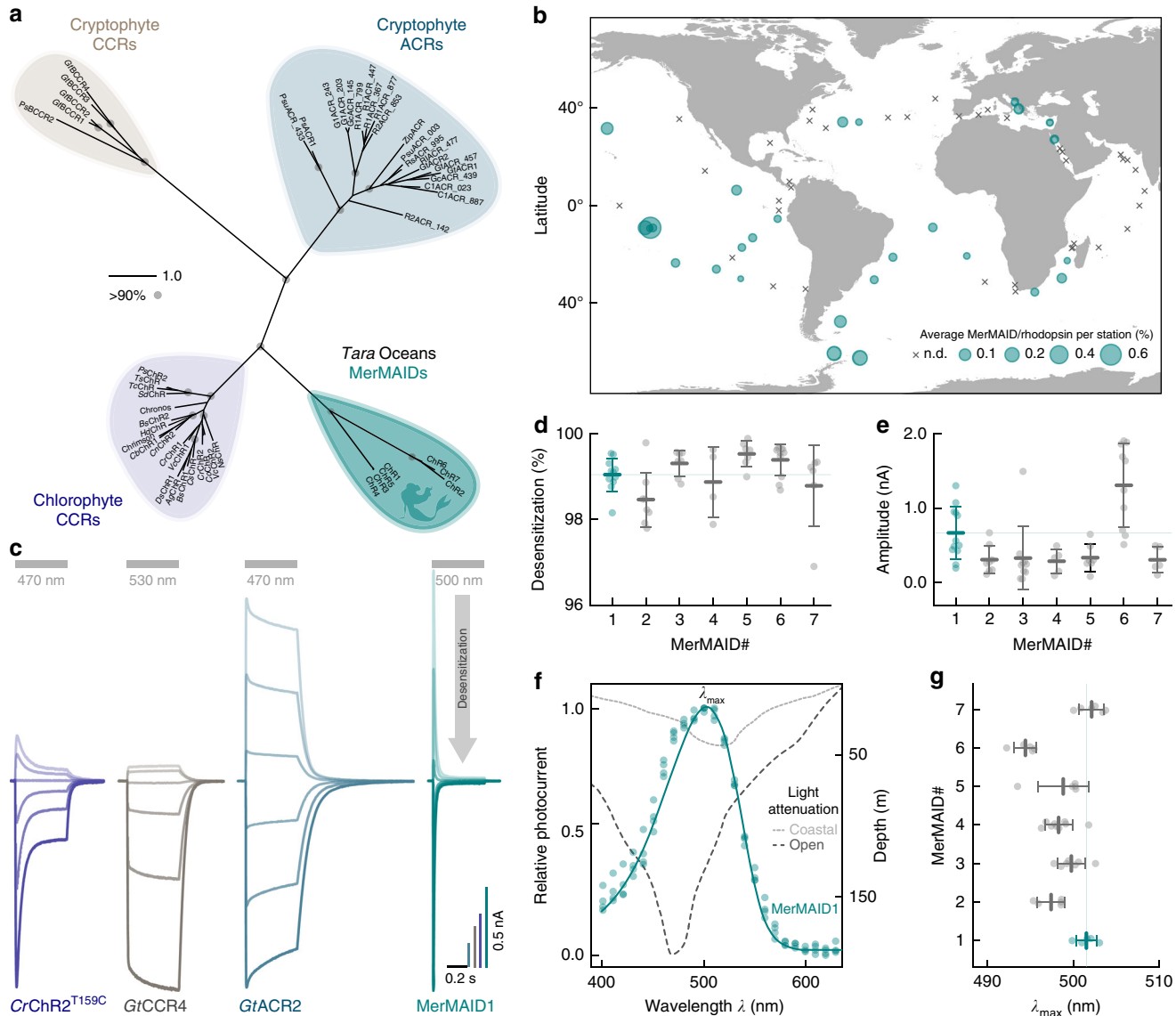

**Fig. 1** Discovery and electrophysiological features of MerMAID.s **a** Unrooted phylogenetic tree of the channelrhodopsin superfamily, with gray circles representing bootstrap values >90%. Scale bar indicates the average number of amino acid substitutions per site. CCR, cation-conducting channelrhodopsin; ACR, anion-conducting channelrhodopsin. An overview of ChRs used to generate the phylogenetic tree can be found in the Supplementary Data 1. **b** Distribution and relative abundance of MerMAIDs in samples from the *Tara* Oceans project. Area of each circle indicates the estimated average abundance of MerMAID-like rhodopsins at different *Tara* Oceans stations. Stations were MerMAIDs were not detected (n.d.) are indicated by crosses. **c** Photocurrent traces of representative members of previously identified ChR families and MerMAIDs, recorded from −60 to +40 mV in steps of 20 or 15 mV (*Gt*CCR4). Gray bars indicate light application at denoted wavelengths. **d**, **e** Desensitization (**d**) and peak current amplitudes (**e**) of all MerMAIDs at −60 mV during continuous illumination with 500 nm light. **f** Normalized action spectrum of MerMAID1. Single measurements are shown as dots ($n = 4$), and the solid line represents fitted data. Dashed lines indicate light penetration depth in coastal and open seawater (adopted from ref. [93]). $\lambda_{max}$, maximum response wavelength; **g** $\lambda_{max}$ for all MerMAIDs. Mean values (thick lines) ± standard deviation (whiskers) are shown, and single-measurement data points are represented as dots. Source data are provided as a Source Data file (**d**–**g**)

mV (Fig. 2h) at different $pH_{e/i}$ values. The peak current recovered with $\tau_{rec} = 1.21 \pm 0.03$ s for MerMAID1 and was unaffected by $pH_e$ or $pH_i$ changes (Fig. 2i, j and Supplementary Fig. 4h, i). Between the different MerMAIDs, $\tau_{rec}$ varied between $1.1 \pm 0.2$ s (MerMAID2) and $6 \pm 1$ s (MerMAID6, Fig. 2j and Supplementary Fig. 4i).

**Accumulation of the late M-state causes desensitization.** To elucidate the desensitization mechanism, recombinant Mer-MAID1 was purified from *Pichia pastoris* and analyzed by UV/vis and vibrational spectroscopy. Steady-state UV/vis absorption spectra of dark-adapted MerMAID1 exhibited a prominent peak

at 502 nm, consistent with the photocurrent action spectra (Fig. 3a). Upon continuous illumination with green light, the 502-nm dark-state absorption peak decreased, while a fine-structured, blue-shifted intermediate with sub-maxima at 346, 364, and 384 nm accumulated in parallel (Fig. 3a, d). Similarly, alkalization converted dark-adapted MerMAID1 into a more blue-shifted, fine-structured UV-absorbing species, consistent with a deprotonated 13-*cis* isomer in the M-state and deprotonated all-*trans* RSB dark state[35] that occurs with a pK value of ~9.8 (Fig. 3b, d and Supplementary Fig. 5d).

Single-turnover voltage-clamp experiments showed a maximum channel conductance 350 μs after ns-pulse laser excitation.

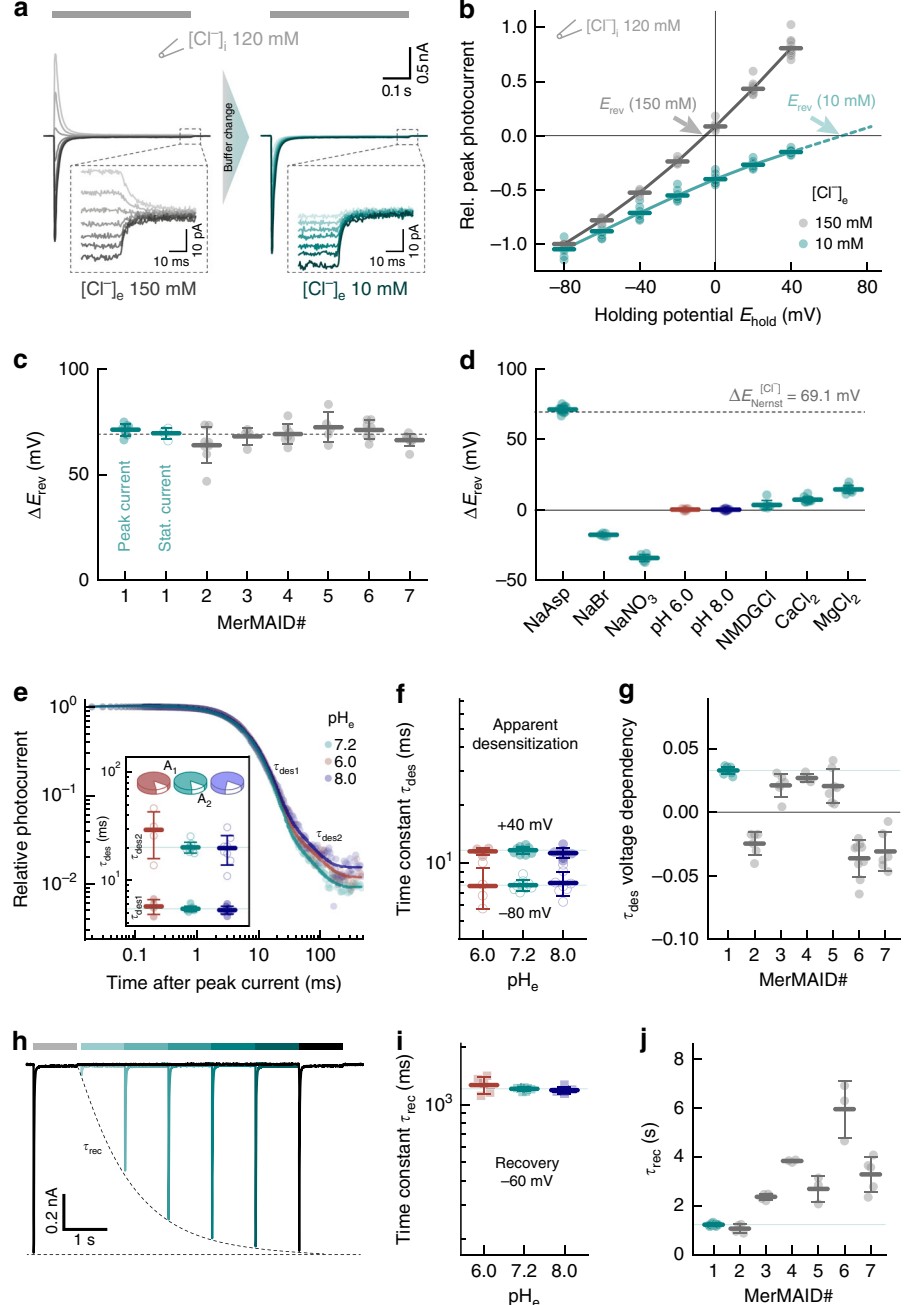

**Fig. 2 Ion selectivity and kinetic properties of MerMAIDs a** Representative photocurrent traces of MerMAID1 elicited with 500 nm light (gray bar) at different membrane potentials (−80 to +40 mV, in 20 mV steps, from bottom to top) before (left, gray) and after extracellular chloride reduction (right, cyan), as indicated. Insets show enlarged views of the remaining stationary photocurrent. **b** Current-voltage relationship of the MerMAID1 peak photocurrent at 150 mM (gray) and 10 mM (cyan) extracellular chloride ([Cl⁻]e). Arrows indicate reversal potentials ($E_{rev}$). **c** Reversal potential shifts ($\Delta E_{rev}$) upon reduction of [Cl⁻]e for peak currents of all MerMAIDs as well as the stationary current of MerMAID1. **d** $\Delta E_{rev}$ values of MerMAID1 upon exchange of external buffer. $\Delta E_{rev}$ of the theoretical Nernst potential for Cl⁻ is indicated as a dashed line (**c**, **d**). **e** Extracellular pH (pHe) dependence of biphasic MerMAID1 desensitization kinetic. Inset shows the time constants and their relative amplitudes to total decay. **f** pHe dependency of the apparent desensitization time constant ($\tau_{des}$) at −80 and +40 mV. **g** Voltage dependency of $\tau_{des}$ for all MerMAIDs in ms/mV. **h** Double-light pulse experiment at −60 mV and pHe 7.2 to determine the peak current recovery time constant ($\tau_{rec}$). **i** pHe dependency of MerMAID1 at −60 mV. **j** Recovery time constants of all MerMAID variants. Mean values (thick lines) ± standard deviation (whiskers) are shown, and single-measurement data points are represented as dots. Source data are provided as a Source Data file (**b–d**, **e–g**, **i**, and **j**)

Channel closing was biphasic, with a dominant fast component and an apparent closing time constant ($\tau_{off}$) of 2.7 ± 0.1 ms (Fig. 3c and Supplementary Fig. 5a). Transient UV/vis absorption spectra (Fig. 3c and Supplementary Fig. 5b, c) revealed an early-decaying (173 ns) K-like photoproduct observed only briefly on our time scale. The evolution-associated difference spectrum

(EADS) of the subsequent L-intermediate is slightly blue shifted and to some extent broadened compared to the dark-state spectrum (Supplementary Fig. 5c), indicating closer proximity of the primary counterion to the RSBH⁺ immediately prior to RSB deprotonation[36]. Within 6 ms, the L-state converted to the M-state, with concomitant deprotonation of the RSB, as indicated by

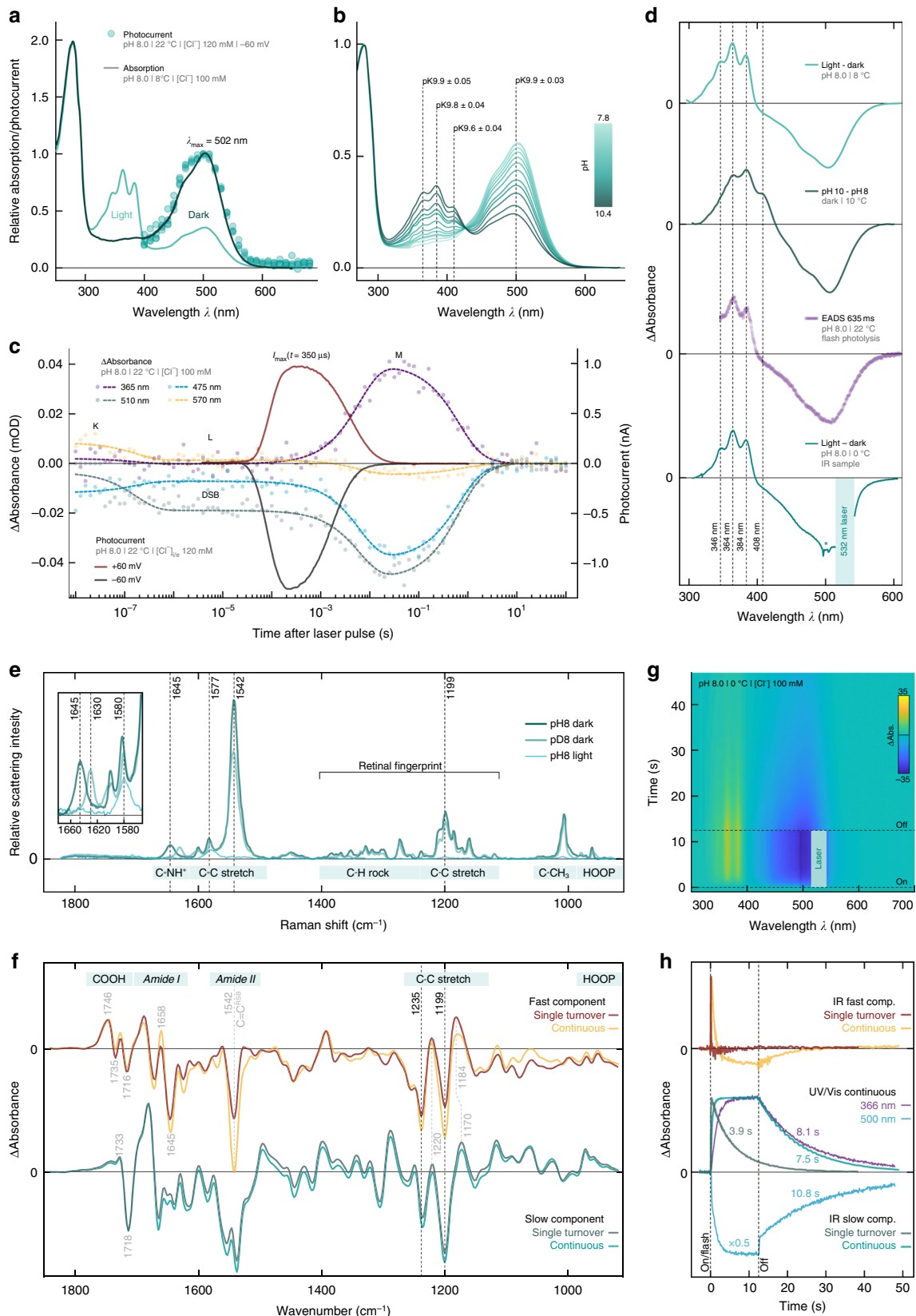

the large blue shift coinciding with channel closure (Fig. 3c and Supplementary Fig. 5a, b). The transient M-state EADS was similarly fine-structured as observed for continuous photoactivation (Fig. 3d), indicating accumulation of the M-state during sustained light exposure.

To assess potential retinal chromophore isomers of Mer-MAID1, we performed resonance Raman (RR) spectroscopy at 80 K. Excitation of dark-adapted MerMAID1 at 488 or 514 nm produced identical RR spectra (Supplementary Fig. 5e), indicating structural homogeneity of the chromophore. The vibrational

**Fig. 3** Spectroscopic characterization of purified MerMAID1. **a** Normalized UV/vis absorption spectra of dark-adapted and illuminated MerMAID1. Filled circles indicate single-measurement action spectra recordings, as shown in Fig. 1f. **b** Normalized UV/vis absorption spectra of MerMAID1 at different pH values, titrated from pH 7.8 to 10.4. The pK values for specific wavelengths are indicated. **c** Transient absorption changes and electrophysiological recordings obtained with single-turnover laser pulse excitation. **d** Fine-structured difference absorption spectra obtained from different experiments. (From top to bottom) light minus dark difference spectra obtained from data shown in **a**, pH-difference spectra calculated from panel b data, evolution-associated difference spectra (EADS) resulting from a global fit of the transient absorption spectra and (bottom) light-minus-dark difference spectrum measured using the FTIR sample shown in **g**. Due to strong laser scattering, a portion of the spectral data is excluded for the FTIR sample, and residual scattering is marked with an asterisk. **e** Resonance Raman spectra of dark-adapted MerMAID1 at pH/D 8 (recorded at 488 nm) as well as cryo-trapped and illuminated protein sample at pH 8 (recorded at 413 nm). Inset: zoomed C=NH$^+$ stretching region. **f** Kinetically decomposed FTIR light-minus-dark absorption of MerMAID1, recorded with single turnover and continuous illumination at 0 °C. Bands marked in gray are discussed in the Supplementary Discussion **g**, Contour plot of transient absorption changes of the sample used in **f** illuminated with a 532 nm continuous laser. **h** Kinetics of the fast and slow FTIR components obtained under single-turnover and continuous illumination conditions, respectively. Kinetics at 366 and 500 nm obtained from the UV/vis spectroscopic measurements shown in **g** are shown for comparison

band pattern in the retinal fingerprint region (1100–1400 cm$^{-1}$) was characteristic of an all-*trans* RSB[37,38] (Fig. 3e). Upon proton/ deuterium exchange, the C=N stretching mode downshifted from 1645 to 1630 cm$^{-1}$ (inset Fig. 3e), indicative of a weakly hydrogen-bonded[39] protonated RSB[37,38]. RR spectra of photo-activated MerMAID1 cryo-trapped in the M-state and excited at 413 nm exhibited a prominent band at 1577 cm$^{-1}$ attributable to a 13-*cis* configuration of the chromophore with deprotonated RSB[40]. RR spectra of dark-adapted MerMAID1 probed with 488 or 413 nm at pH 10 were similar to spectra of dark- and light-adapted MerMAID1 at pH 8 (Supplementary Fig. 5e). Notably, strong bands in the C=C stretching region at ca. 1540 and 1577 cm$^{-1}$ indicated a mixture of the protonated and deprotonated all-*trans* RSB of the dark state at high pH. Thus, RR spectra confirmed that MerMAID1 undergoes all-*trans* to 13-*cis* retinal isomerization and deprotonation of the RSB during illumination and may deprotonate at high pH in the dark.

Time-resolved Fourier-transform infrared (FTIR) spectra were collected at pH 8.0 and 0 °C to examine the light-driven molecular processes of MerMAID1 under single-turnover conditions and continuous illumination (Fig. 3f, h), with parallel UV/vis observation of M-state formation (Fig. 3g, h). Kinetic decomposition of light-dark FTIR difference spectra revealed highly similar fast and slow spectral components for both single-turnover and continuous illumination, respectively (Fig. 3f). At both conditions, the slow FTIR component relaxed to the dark state mono-exponentially, within seconds (Fig. 3h) and was assigned to the late M-state that accumulated with continuous illumination (Fig. 3g, h) without formation of other photoproducts. This assignment was supported by data for the retinal fingerprint region that - similar to the RR data - indicated all-*trans* to 13-*cis* retinal isomerization as the only photoreaction based on the negative bands at 1235(−) and 1199(−) cm$^{-1}$. The fast FTIR spectral components resembled the short-lived conducting L-state preceding the late M-intermediate, as inferred from the comparable decay kinetics (see Supplemental Discussion).

**A conserved cysteine is critical for the desensitization**. Site-directed mutagenesis guided by MD simulations and probed by electrophysiological recordings were conducted to further examine the molecular mechanism for the intense desensitization of the MerMAIDs. For MD simulations, a MerMAID1 homology model was constructed based on the iC++ crystal structure[41] and embedded in a phospholipid bilayer (Fig. 4a and Supplementary Fig. 6a). The D210, E44, W80, and Y48 side chains located near the protonated Schiff base (Fig. 4a, c) maintained their relative positions during a 100-ns MD simulation. The orientation and distances of these residues changed only slightly with inflowing water (Supplementary Fig. 6d–f). Possible ion translocation pathways were calculated using

MOLEonline[42]. Figure 4a, b shows the most likely ion pathway based on surface charge considerations (Supplementary Fig. 6b, c). Extracellularly, MerMAID1 is accessible via a narrowing tunnel that is disrupted by W80, D210, and the RSB (Fig. 4a, b). Intracellularly, another ion pathway is formed leading from the protein surface almost to the Schiff base, disconnected only by a short hydrophobic barrier. In our model, the carboxylic residues of the active-site complex (E44 and D210) were deprotonated based on pK$_a$ calculations (Fig. 4c, pK$_a$ < 5.5). D210, which acts as closest counterion (2.6 Å) to the Schiff base nitrogen, primarily stabilizes the RSB proton (Fig. 4c). The carboxyl group of D210 also interacts with S79, Y48, and E44 via two water molecules, whereas E44 hydrogen bonds directly to Y48 and is linked to the backbone oxygen of D210 via another water molecule (Fig. 4c). When the counterion D210 is neutralized (D210N), photocurrents are drastically reduced (Fig. 4d, e), $\lambda_{max}$ was 16 nm red-shifted (Fig. 4f), and the recovery kinetics decelerated markedly (Fig. 4h). Elimination of the more distant E44 via an E44Q mutation caused only a 3 nm bathochromic action-spectrum shift (Fig. 4f), indicative for a protonated E44 in wild-type MerMAID1 unlike predicted from our model structure pK$_a$ calculations. Moreover, the E44Q mutation increased the photocurrent amplitudes (Fig. 4e), decelerated desensitization by a factor of 10 (Fig. 4d, g) and slightly reduced the extent of desensitization (Fig. 4i). Replacement of both acidic residues (E44Q-D210N) only halved photocurrent amplitudes (Fig. 4e) and shifted $\lambda_{max}$ to 513 ± 1 nm (Fig. 4f), suggesting rearrangement of the hydrogen bond network around the RSB. Desensitization remained strong (Fig. 4i), but the kinetics slowed, similar to the E44Q mutation alone (Fig. 4g).

Neutralization of E44 increased the stationary current only slightly (Fig. 4i), whereas we identified C84 (the *Cr*ChR2 C128 homolog) as a crucial determinant of the inactivation process. The C84T mutant exhibited a decreased peak current amplitude but markedly increased stationary photocurrent (Fig. 4d, e), resulting in only 65 ± 5% desensitization (Fig. 4d, i) and minimally altered desensitization kinetics (Fig. 4d, g). In contrast, we observed no peak current recovery within a time period of 200 s. As suggested by our model structure and the pronounced 17 nm blue-shifted $\lambda_{max}$ (Fig. 4f), C84 is located near the retinal polyene chain and the C13 methyl group (Fig. 4a, c). In MerMAIDs, this cysteine cannot serve as a link between helices 3 and 4 as discussed for bacteriorhodopsin[43,44] and CCRs[45,46] due to the absence of a hydrogen-bonding partner in helix 4 (Supplementary Fig. 2).

**Temporally precise neuronal silencing using MerMAIDs**. Finally, we evaluated the utility of MerMAIDs as optogenetic tools for inhibiting neuronal activity. As MerMAID6 exhibited

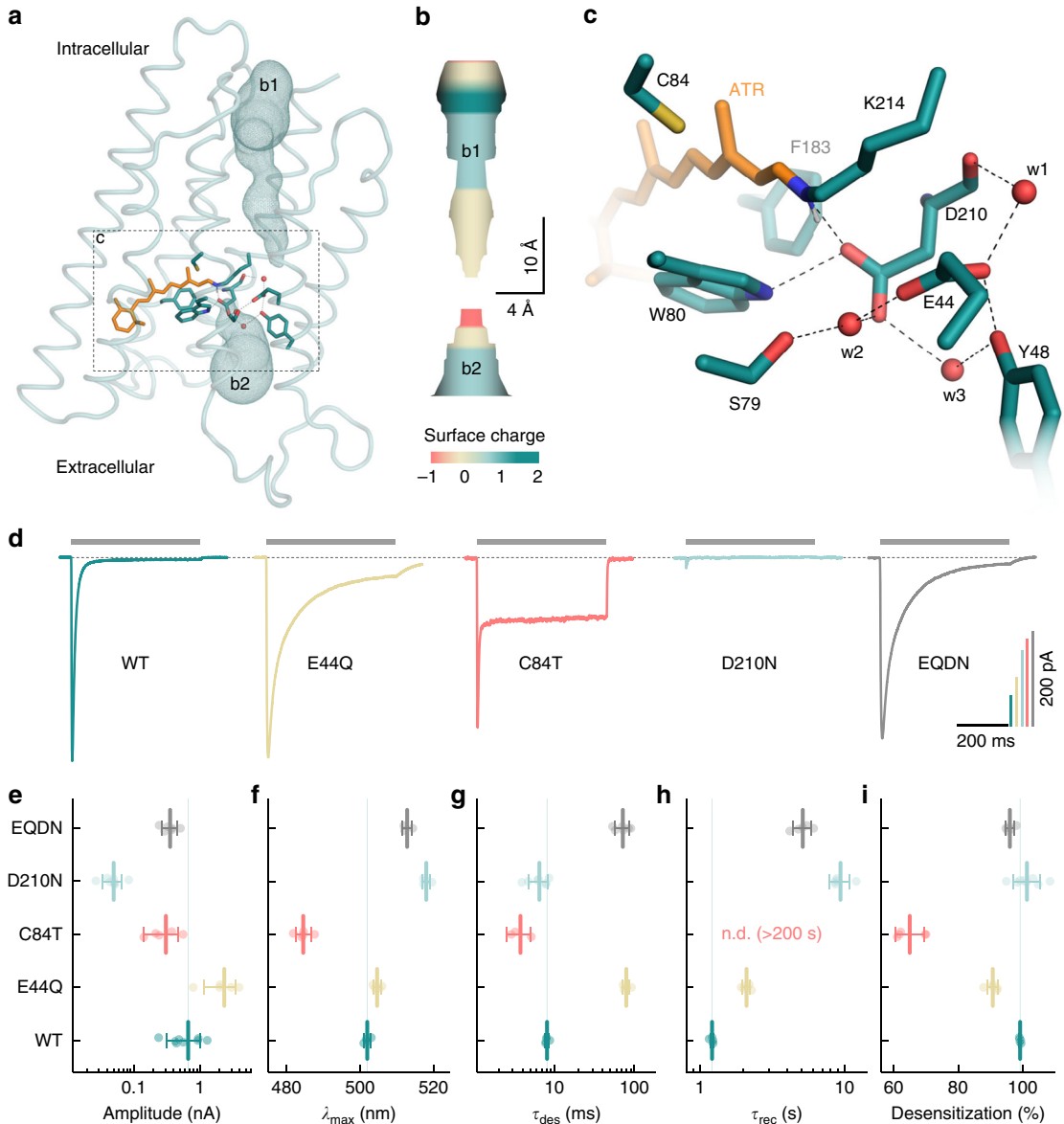

**Fig. 4** MD simulations and mutational analysis of MerMAID1. **a** Overview of the MD simulation homology model of MerMAID1 in the dark. The predicted ion permeation pathway is shown as mesh (b1, b2), and ribbons represent the protein backbone. **b** Electrostatic surface potential of the predicted chloride permeation pathway. **c** Detailed view of the active-site residues, with amino acids shown as cyan sticks and the all-*trans* retinal (ATR) in orange. Red spheres denote water molecules that remained stable during MD simulation. **d** Representative photocurrent traces of wild-type (WT) MerMAID1 and selected MerMAID1 mutants recorded at −60 mV. Photocurrent amplitudes (**e**), $\lambda_{max}$ (**f**), apparent $\tau_{des}$ of the peak current (**g**), recovery time constant, $\tau_{rec}$ (**h**), and extent of desensitization (**i**) of WT MerMAID1 and indicated mutants. Mean values (thick lines) ± standard deviation (whiskers) are shown, and single-measurement data points are represented as dots. Source data are provided as a Source Data file (**e–i**)

the highest photocurrent in HEK cells (Fig. 1e), we generated a Citrine-labeled MerMAID6 variant and co-expressed it with mCerulean as a volume marker. MerMAID6-Citrine expression was readily detected in CA1 pyramidal neurons of hippocampal slice cultures 4–5 days after single-cell electroporation. We observed membrane-localized MerMAID6 expression, with some fraction of the protein displaying a speckled cellular distribution (Fig. 5a). However, illumination triggered high transmembrane photocurrents with biophysical properties similar to those observed in HEK cells (Supplementary Fig. 8). The large, transient photocurrents observed in neurons led us to hypothesize that MerMAID6 could be used to block single action potentials (APs) with high temporal precision and without affecting subsequent APs in the presence of light. We first

injected a depolarizing current ramp into the soma to precisely determine the rheobase for AP firing in the dark. A 10 ms light pulse synchronized with the first AP that occurred during darkness eliminated generation of this AP (Fig. 5b). We then applied a 500 ms light pulse synchronized to the time of onset of the first AP lasting throughout the remainder of the current ramp (Fig. 5c) or a depolarizing current step (Fig. 5d), MerMAID6 suppressed generation of the first AP, without affecting the following ones due to rapid accumulation of the desensitized and non-conducting state during extended illumination. Similarly, selective inhibition of a single AP was achieved with MerMAID1 (Supplementary Fig. 7), demonstrating that photoactivated MerMAID1 and MerMAID6 provide efficient and temporally precise inhibition of neuronal activity.

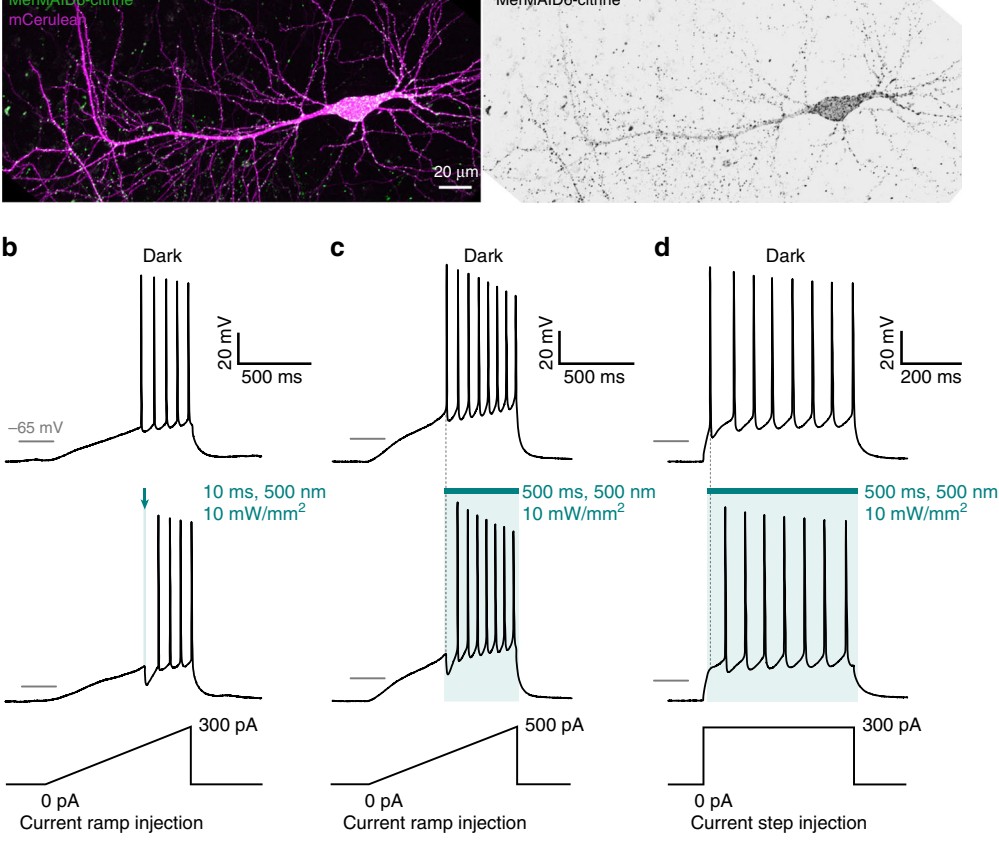

**Fig. 5** Neuronal application of MerMAID6 as optogenetic silencer. **a** CA1 pyramidal neuron expressing MerMAID6-Citrine (green) 5 days after electroporation (stitched maximum intensity projections of two-photon images). mCerulean (magenta) was co-electroporated to visualize neuronal morphology (left). Fluorescence intensity shown as inverted gray values (right). **b**, **c** Voltage traces in response to depolarizing current ramps injected into MerMAID6-expressing CA1 pyramidal cells. Illumination with green light (500 nm, 10 mW/mm2) for a brief (10 ms, **b**) or longer (500 ms, **c**) time period blocked single spikes. Light onset preceded action potential onset (measured in the dark condition) by 5 ms. **d** Same as **c** but a depolarizing current step of 300 pA was injected into the neuron instead of a current ramp

## Discussion

We extensively characterized the biophysical properties of the MerMAIDs, a family of ChRs identified from metagenomic data. All MerMAIDs share similar activity maxima optimal for sensing light in moderate-depth seawater. Similar to other recently discovered natural ACRs[23], MerMAIDs selectively conduct anions. Distinct from all other ChRs, MerMAIDs exhibit almost complete desensitization during exposure to continuous bright light. However, the environmental advantage of near-complete desensitization compared with non-inactivating ACRs is unclear.

After photon absorption, the MerMAID1 chromophore isomerizes from all-*trans* to 13-*cis*, as demonstrated by RR and FTIR spectroscopy. We hypothesize that the RSBH$^+$ dipole changes orientation and distance with respect to the nearby D210, as evidenced by formation of the L-intermediate, analogous to bacteriorhodopsin[36]. Following the K→L transition, the L state remains UV/vis spectroscopically unchanged over almost three temporal orders of magnitude, while channel opening proceeds during the long-lived L-state. Accordingly, and in line with FTIR data (see Supplemental Discussion), only minimal protein backbone changes involving residues in the vicinity of the RSB can take place during formation of the conducting state. Maximum channel conductivity is reached within 350 μs. Channel closing proceeds concurrently with RSB deprotonation, leading to M-intermediate formation, similar to cryptophyte ACRs[24,47] but in

contrast to chlorophyte CCRs, in which M-state formation precedes channel opening[48,49]. These observations suggest that the positively charged protonated RSB is part of the chloride-conducting pathway, consistent with the calculated ion permeation pathway along the counterion complex, similar to crystal structures of *Guillardia theta* ACR1 (*Gt*ACR1)[41,50]. Chloride flux in MerMAID1 is interrupted by lack of negative charge attraction following deprotonation of the Schiff base. In the final photocycle step, MerMAID1 structurally rearranges, the RSB reprotonates and reisomerizes to all-*trans* during recovery of the initial dark state that is fully repopulated within seconds.

The observation that photocurrent kinetics were not affected by intra- or extracellular pH changes suggests that the RSB proton remains within the central active-site complex during the photocycle, as recently reported for heliorhodopsins[31]. D210 is the primary counterion of the RSB in MerMAID1, but both carboxylic residues (E44 and D210) participate in de- and reprotonation of the MerMAID1 chromophore, as neutralization of either one or both residues affects formation of the conductive or desensitized state. However, retention of function of the E44Q-D210N double mutant suggests the possibility of alternative proton acceptor and donor sites.

The unique desensitization of the MerMAIDs can be explained by the accumulation of the blue-shifted M-intermediate during constant photoactivation. Because the non-conducting M-

intermediate is formed within milliseconds and decays only within hundreds of milliseconds, the current declines to 1 % in continuous light. This mechanism is consistent with an M-state that cannot be photochemically converted back to the dark state, which is the case for MerMAID1 as demonstrated by electrophysiological and IR spectroscopic data (Supplementary Fig. 5g–i), while the underlying mechanism is not understood. As discussed in previous reports, decline of CrChR2 photocurrents upon continuous illumination is due to both, the accumulation of late non-conducting photocycle intermediates and population of a parallel (syn-) photocycle with an only weakly conducting open state[7–9]. The accumulation of a late photocycle intermediate is the dominant mechanism in MerMAID1, as demonstrated by FTIR spectroscopy; no parallel photocycle is needed to explain the strong inactivation.

We found that replacing C84 in MerMAID1 decreases the peak photocurrent but increases stationary photocurrents, thus reducing the extent of desensitization, possibly due to either a prolonged L-state or a shortened recovery from the M- to the initial dark state. However, in the C84T mutant, the desensitization kinetic was found to be accelerated and recovery to the dark state was slowed down. Hence, stationary photocurrents might be potentially a cause of a branched photocycle instead, in which a second photoactive closed state can be populated from the initial dark state by photon absorption, as discussed for various CCRs[7,8,51–53]. As recently substantiated experimentally for CrChR2, the retinal in both parallel photocycles differ, adopting either a syn or an anti conformation[10]. Conceivably, C84 could suppress a C=N anti to syn isomerization in MerMAID1 and therefore prevent population of parallel syn-photocycle that could account for a second conducting state and stationary photocurrents. However, future studies are necessary to prove presence of various retinal isomers during the photocycle of MerMAID1 C84T.

Another unusual feature of MerMAID1 are the fine-structured absorption spectra of both the deprotonated all-trans RSB in the dark at alkaline pH and the 13-cis retinal of the M-state. Such unusual spectra have been reported for other microbial rhodopsins after retro-retinal formation upon reduction with borohydride[54] or hydrolysis of the RSB[55]. In both cases, the UV fine structure results from immobility of the deprotonated chromophore, which is typically more pronounced at deep temperatures[56,57]. Alkalization-induced fine-structured spectra were reported for eACRs[41] and wild-type and mutant nACRs[47] and suggested to result from RSB hydrolysis[41] or protein denaturation[58]. However, in GtACR1, the covalent bond between the retinal and the Schiff base–forming lysine is not broken at high pH. Instead, the RSB deprotonates and adopts an M-like configuration[59]. RR spectra of MerMAID1 at pH 10 (Supplementary Fig. 5e) possibly suggested that the retinal is similarly deprotonated and adopts a rigid configuration in all-trans instead of 13-cis, as in the M-state.

MerMAID1 and MerMAID6 effectively inhibited neuronal activity with high temporal precision. Due to the unique desensitization of the MerMAIDs under continuous illumination, single APs can be blocked at the onset of illumination without affecting subsequent neuronal spiking. Hence, MerMAIDs could serve as transient optogenetic silencers to inhibit individual APs with high precision in combination with subsequent imaging of spectrally overlapping reporters of neuronal activity. MerMAIDs would thus facilitate continuous monitoring of neuronal activity subsequent to short-duration inhibition at the same wavelength.

The identification of the MerMAID ChR family fortifies the value of metagenomic data for the discovery of photoreceptor proteins potentially applicable as optogenetic tools. The initial indepth characterization of MerMAIDs will foster the generation of ChRs with biophysical properties and lead to deeper understanding of the working principles of rhodopsins.

## Methods

**ChR identification and metagenomics data analysis**. ChR variants were identified using full-length CrChR1 and CrChR2 amino acid sequences (GenBank IDs: AF461397.1 and AF385748.1, respectively) as queries for tblastn 2.6.0 analysis[60] against a database of contigs assembled from the Tara Oceans metagenomic datasets of bacterial[61], viral[62], and girus[63] samples. The assemblies were generated as described elsewhere[64].

MerMAID abundance in the marine environment was estimated using the Ocean Gene Atlas[65] after mining the Ocean Microbial Reference Gene Catalog[61]. A collection of representative microbial rhodopsin protein sequences from distinct subfamilies containing the MerMAIDs was aligned using the MAFFT online server (ver. 7)[66]. The alignment was used to generate a Hidden Markov Model (HMM) using hmmbuild from the HMMER 3.1b2 suite[67]. The HMM served as the query in the Ocean Gene Atlas[65] or an HMMER-based search with default parameters against the Ocean Microbial Reference Gene Catalog. The Ocean Gene Atlas results for abundances and homologs were stored locally for further analysis.

Protein homologs from the Ocean Gene Atlas and MerMAIDs were pooled and aligned using the MAFFT web server. MAFFT multiple sequence alignment was used to identify those homologs phylogenetically closer to the MerMAIDs and were tagged as MerMAID-like. Ocean Gene Atlas abundance data were parsed using a custom R script to calculate the ratio of ACR-like proteins to total rhodopsins in each Tara Oceans sample. The MerMAID-like/total rhodopsin ratio was coupled with environmental metadata from the Tara Oceans samples to generate depth profiles and distribution maps using the R packages maps[68], ggplot2[69], and ggalt[70].

The phylogenetic tree was generated using phylogeny.fr[71] and the sequence alignment using Clustal X[72]. The sequence alignment was visualized using the ENDscript 2 web server[73], and the alignment was cropped to include the transmembrane regions of selected ChRs.

**Molecular biology and protein purification**. Human/mouse codon-optimized sequences encoding MerMAIDs were synthesized (GenScript, Piscataway, NJ) and cloned into the p-mCherry-C1 vector using NheI and AgeI restriction sites (FastDigest, Thermo Fisher Scientific, Waltham, MA) for electrophysiologic recordings in HEK293 cells. Due to incomplete metagenomic data, a methionine was added as start codon for MerMAID1 and MerMAID4. Site-directed mutagenesis of the MerMAID1 gene was performed using Pfu polymerase (Agilent Technologies, Santa Clara, CA) using the following oligonucleotides: GTGTCCGGCGTGCAG TTCATC (E44Q_fwd), GATGAACTGCACGCCGGACAC (E44Q_rev), CTGGC CACCACCCCAATCATC (C84T_fwd), GATGATTGGGGTGGTGGCCAG (C84T_rev), GTGATCGGCAACGTGATCAGCAAG (D210N_fwd) and CTTGCTGATCACGTTGCCGATCAC (D210_rev). MerMAID1 and MerMAID6 cDNAs were subcloned into neuron-specific expression vectors (pAAV backbone, human synapsin promoter) in frame with Citrine cDNA using Gibson assembly[74]. For expression in Pichia pastoris, the MerMAID1 gene was subcloned with a C-terminal TEV protease restriction site and a 6× His-Tag into the pPiCZ vector (Invitrogen, Carlsbad, CA). Zeocin™-resistant positive clones were selected from electroporation-transformed yeast cells. Expression of MerMAID1 in precultured cells was induced with 2.5% methanol in presence of 5 μM all-trans retinal for 24 h. Cells were harvested by centrifugation and resuspended in breaking buffer (50 mM NaPO$_4$, 1 mM EDTA, 1 mM PMSF, 5% glycerol [pH 7.4]) and disrupted by high pressure using a French press (G. Heinemann Ultraschall und Labortechnik, Schwäbisch Gmünd, Germany). The membrane fraction was collected, homogenized, and solubilized overnight at 4 °C in 100 mM NaCl, 20 mM Tris-HCl, 20 mM imidazole, 1 mM PMSF, 5 μM all-trans retinal, and 1% (w/v) dodecyl maltoside (DDM). Recombinant rhodopsin was purified by affinity chromatography (HisTrap™ FF Crude column, GE Healthcare Life Science, Chicago, IL) and gel filtration (HiPrep™ 26/10 desalting, GE Healthcare Life Science). Before elution, an additional washing step with buffer containing 50 mM imidazole was performed. Purified protein was concentrated and stored in 100 mM NaCl, 20 mM Tris-HCl (pH 8), and 0.05% DDM.

**Electrophysiology in HEK-293 cells**. HEK-293 cells (ECACC 85120602, Sigma-Aldrich, Munich, Germany) were cultured and electrophysiologic experiments were performed as described elsewhere[22,75]. In detail, cells were supplemented with 1 μM all-trans retinal, seeded at a density of $1 \times 10^5$/ml on poly-D-lysine–coated coverslips, and transiently transfected using Fugene HD (Promega, Madison, WI). At 1–2 days post-transfection, whole-cell patch-clamp recordings were performed at 24 °C. The resistance of fire-polished patch pipettes was 1.5–2.5 MΩ, and a 140 mM NaCl agar bridge served as the reference electrode. Membrane resistance was generally ≥1 GΩ, and the access resistance was <10 MΩ. Signals were amplified (AxoPatch200B), digitized (DigiData400), and acquired using Clampex 10.4 (all from Molecular Devices, Sunnyvale, CA). Light from a Polychrome V (TILL Photonics, Planegg, Germany) with 7 nm bandwidth was channeled into an Axiovert 100 microscope (Carl Zeiss, Jena, Germany) controlled via a programmable shutter system (VS25 and VCM-D1; Vincent Associates, Rochester, NY). Light intensity was measured in the sample plane using a calibrated optometer

(P9710; Gigahertz Optik, Türkenfeld, Germany) and calculated for the illuminated field of the W Plan-Apochromat ×40/1.0 DIC objective (0.066 mm2, Carl Zeiss). Final buffer osmolarity was set with glucose to 320 mOsm (extracellular) or 290 mOsm (intracellular), and the pH was adjusted using N-methyl-D-glucamine or citric acid. Liquid junction potentials (Supplementary Table 1) were calculated (Clampex 10.4) and corrected. For ion selectivity measurements, extracellular buffers (Supplementary Table 1) were exchanged in random order by adding at least 3 ml to the measuring chamber (volume ~0.5 ml), while the fluid level was kept constant using an MPCU bath handler (Lorentz Messgerätebau, Katlenburg-Lindau, Germany). MerMAID photocurrents were induced for 500 ms and recorded between −80 and +40 mV in 20-mV steps. Low-intensity light between 390 and 680 nm was applied in 10-nm steps for 10 ms at −60 mV to generate action spectra. Equal photon irradiance at all wavelengths was achieved using a motorized neutral-density filter wheel (Newport, Irvine, CA) in the light path, controlled by custom software written in LabVIEW (National Instruments, Austin, TX). For light titration experiments, photocurrents were induced for 2 s at −60 mV, and light was attenuated using ND filters (SCHOTT, Mainz, Germany) inserted into the light path using a motorized, software-controlled filter wheel (FW212C, Thorlabs, Newton, NJ). Single-turnover experiments were performed with the above mentioned setup described elsewhere[10,76]. An Opolette HE 355 LD Nd:YAG laser/OPO system (OPOTEK, Carlsbad,CA) served as pulsed laser light source.

**UV-Vis Spectroscopy.** Steady-state absorption spectra were recorded using a Cary 300 UV/vis spectrophotometer (Varian Inc., Palo Alto, USA) or UV-2600 UV/vis spectrophotometer (Shimadzu, Kyōto, Japan) at a spectral resolution of 1 nm in buffer containing 100 mM NaCl, 20 mM Tris, and 0.05% DDM (pH 8). Data was collected using UVProbe v2.34 software (Shimadzu) or Varian UV v3.0 software (Varian). Light-adapted absorption spectra were acquired by illuminating the sample with a 530 nm LED with a 520 ± 15 nm filter. For pH titration experiments, small volumes of 1 M NaOH were added to samples in titration buffer (100 mM NaCl, 10 mM BTP, 10 mM CAPS, 0.05% DDM [pH 7.5]). The pH was measured using pH microelectrodes (SI Analytics, Mainz, Germany). Single-turnover transient absorption spectroscopic measurements were performed as described elsewhere[24] at 22 °C using a modified LKS.60 flash-photolysis system (Applied Photophysics Ltd., Leatherhead, UK). For sample excitation, the laser pulse was tuned to 500 nm using a optical parametric oscillator (MagicPrism, OPOTEK), which was pumped with the third harmonic of a Nd:YAG laser (BrilliantB, Quantel, Les Ulis, France). The laser energy was adjusted to 5 mJ/shot and pulse duration of 10 ns. A 150-W xenon lamp (Osram, München, Germany) was used to monitor changes in absorption. Transient spectra were recorded in multi-wavelength data sets at a resolution of 0.4 nm using an Andor iStar ICCD camera (DH734; Andor Technology Ltd, Belfast, Ireland). Spectra were recorded at 101 different time points between 10 ns and 100 s (10 points per decade, iso-logarithmically) with custom software written in Visual C++. To ensure complete recovery of the dark state, samples were kept in the dark for 120 s before the subsequent recording. For the transient absorption spectra shown in Fig. 3h, FTIR samples were used. Spectra were recorded using an Ocean FX array detector (Ocean Optics, Largo, FL) with a spectral resolution of 2.4 nm and integration time of 50 ms. Data was collected using custom written in C#. Samples were illuminated using a 50 mW continuous-wave LASER emitting 532-nm light (no. 37028, Edmund Optics, York, UK).

**FTIR spectroscopy.** To prepare samples for FTIR, 10 μl of initial protein solution (>20 mg/ml, 100 mM NaCl, 20 mM Tris, 0.05% DDM [pH 8]) was dried stepwise on a BaF2 window under a stream of dry air and subsequently rehydrated. Samples were then sealed with a second BaF2 window. To ensure constant sample thickness, a 3-μm PTFE spacer was placed between the windows. For deuteration, the protein solution was washed at least five times with deuterium buffer (100 mM NaCl, 20 mM Tris, 0.05% DDM [pD 8]) using Centricon filters and subsequently illuminated using white light to improve intramolecular deuteration. FTIR spectra were acquired at 0 °C using a Vertex 80 v FTIR spectrometer (Bruker Optics, Karlsruhe, Germany), equipped with a liquid nitrogen–cooled MCT detector (Kolmar Technologies, Newburyport, MA), using OPUS 7.5 software (Bruker). The spectrometer was operated in rapid scan mode with a data acquisition rate of 300 kHz and spectral resolution of 8 cm$^{-1}$. An optical cutoff filter at 1850 cm$^{-1}$ was used in the beamline. After at least 60 min for equilibration, samples were continuously illuminated using a set of green-light LEDs with an emission maximum of 520 nm. Additional UV light application was performed with a set of 362 nm LEDs. Single-turnover illumination was performed using a 10-Hz pulsed Nd:YAG Powerlite 9010 LASER as the pump source for a Horizon II optical parametric oscillator (Continuum, San Jose, CA) set to 530 nm. The pulse width of the setup was approximately 5 ± 2 ns, with an energy output of approximately 60 mJ. The time resolution was 6 ms (achieved by operation in forward-backward mode and splitting of the interferogram).

**RR spectroscopy.** RR spectra were acquired with excitation lines of an Ar$^+$ (514 nm, 488 nm) and Kr$^+$ laser (413 nm) (Coherent, Santa Clara CA). Raman signals were detected in a backscattering configuration (180°) using a confocal LabRamHR

spectrometer (Horiba, Villeneuve, France) equipped with a liquid nitrogen–cooled CCD detector. Data was collected with the LabSpec Spectroscopy Suite software (Horiba). The spectral resolution was approximately 2 cm$^{-1}$. Typical total accumulation time and laser power at the sample were 30 min and 1 mW, respectively. Low-temperature measurements at 80 K were carried out with a Linkam cryostat (Linkam Scientific Instruments, Surrey, UK). Samples were inserted into the cell under dimmed red light in order to avoid photoactivation before freezing.

**MD simulations.** Classical MD simulations were prepared based on a SWISS homology model[77] of MerMAID1 on iC++ at pH 8.5 (PDB 6CSN). The iC++ structure was chosen as template as it showed the best combined quality features for structural prediction (best coverage and QMQE [0.61 together with PDB 6CSM], 2nd best QSQE [0.27 vs. 0.28 with PDB 4YZI], and 3rd best identity [32.69% vs. 35.61% with PDB 6EID]). The model was prepared using CHARMM-GUI[78] for the resting state of MerMAID1 with standard protonation for all amino acids. The MerMAID1 monomer was embedded inside a 60 × 60 Å, homogeneous, 1,2-dimyristoyl-sn-glycero-3-phosphocholine bilayer membrane and solvated using a TIP3 water box, adding 10 Å to both the top and bottom of the protein. Systems were simulated under NPT conditions using a 2 fs time step, a 303.15 K heat bath, the particle-mesh Ewald method for long-range electrostatics, and the CHARMM36 force field[79]. pK$_a$ calculations for all titratable amino acids of MerMAID1 were performed using APBS[80] in a conformational space of three pH-adapted conformations (PACs) and the Monte Carlo procedure of Karlsberg2+[81,82] to sample all residues. PACs were created using Karlsberg2+ in a self-consistent cycle including adjustment of protonation patterns of titratable amino acids and salt bridge opening according to pH −10, 7, or 20. To calculate pK$_a$ values for Mer-MAID1 MD frames, only the holoprotein structure was used. Lipids and water molecules were substituted with continuum solvation. Ion channels were predicted using MOLEonline[42].

**Neuronal recordings and two-photon microscopy.** Organotypic slice cultures of rat hippocampus were prepared as described[83] and transfected by single-cell electroporation[84] after 14–16 days in vitro (DIV). Plasmids were each diluted to 1 ng/μl in K-gluconate–based solution consisting of (in mM): 135 K-gluconate, 4 MgCl2, 4 Na2-ATP, 0.4 Na-GTP, 10 Na2-phosphocreatine, 3 ascorbate, 0.02 Alexa Fluor 594, and 10 HEPES (pH 7.2). An Axoporator 800 A (Molecular Devices) was used to deliver 50 hyperpolarizing pulses (−12 mV, 0.5 ms) at 50 Hz. At DIV 19–21, targeted patch-clamp recordings of transfected neurons were performed under visual guidance using a BX 51WI microscope (Olympus, Shinjuku, Japan) equipped with Dodt-gradient contrast and a Double IPA integrated patch amplifier controlled with SutterPatch 2.0 software (Sutter Instrument, Novato, CA), also used for data acquisition. Patch pipettes with a tip resistance of 3–4 MΩ were filled with intracellular solution consisting of (in mM): 135 K-gluconate, 4 MgCl2, 4 Na2-ATP, 0.4 Na-GTP, 10 Na2-phosphocreatine, 3 ascorbate, 0.2 EGTA, and 10 HEPES (pH 7.2). Artificial cerebrospinal fluid (ACSF) consisted of (in mM): 135 NaCl, 2.5 KCl, 2 CaCl2, 1 MgCl2, 10 Na-HEPES, 12.5 D-glucose, 1.25 NaH2PO4 (pH 7.4). Synaptic currents were blocked with 10 μM CPPene, 10 μM NBQX, and 100 μM picrotoxin (Tocris, Bristol, UK). Measurements were corrected for a liquid junction potential of −14.5 mV. A 16-channel pE-4000 LED light engine (CoolLED, Andover, UK) was used for epifluorescence excitation and delivery of light pulses for optogenetic stimulation (ranging from 385–635 nm). Light intensity was measured in the object plane with a 1918 R power meter equipped with a calibrated 818 ST2 UV/D detector (Newport) and divided by the illuminated field (0.134 mm2) of the LUMPLFLN 60XW objective (Olympus).

Neurons in organotypic slice cultures were imaged with two-photon microscopy to characterize their morphology and the subcellular localization of citrine-labeled MerMAID-ChRs. The custom-built two-photon imaging setup was based on an Olympus BX-51WI upright microscope upgraded with a multiphoton imaging package (DF-Scope, Sutter Instrument), and controlled by ScanImage 2017b (Vidrio Technologies, Ashburn, VA), also used for data collection. Fluorescence was detected through the objective (NIR Apo 40XW, Nikon, Minato, Japan) using GaAsP-PMTs (Hamamatsu Photonics, Hamamatsu, Japan). A tunable Ti:Sapphire laser (Chameleon Vision-S, Coherent) was set to 810 nm to excite mCerulean, and a high power femtosecond fiber laser (Fidelity-2, Coherent, Santa Clara, CA) was used to excite citrine at 1070 nm.

Animal procedures were in accordance with the guidelines of local authorities and directive 2010/63/EU.

**Data analysis and statistical methods.** Clampfit 10.4 (Molecular Devices) and Origin 2017 (OriginLab, Northampton, MA) were used for analysis of HEK293 electrophysiological recordings. Peak currents were used for analysis of most biophysical properties. The current of the last 50 ms of the illumination period was averaged to determine stationary current amplitude. Reversal potentials were determined based on linear fit of the two data points crossing 0 pA or linear extrapolation from 0 pA most adjacent two data points of a measurement series. Action spectra were normalized to the maximum response and fitted with a three-parametric Weibull function to determine the maximum response wavelength (λ$_{max}$). Kinetic time constants were determined by mono or bi-exponential fits. For displaying reasons electrophysiological recording data points were reduced.

Single turnover UV/vis absorption measurements were averaged over 15 cycles. Primary data analysis was performed using MATLAB R2016b (The MathWorks, Natick, MA) to calculate difference spectra and reconstruct three-dimensional spectra. Glotaran 1.5.1[85,86] was used for global analysis of the spectral datasets. Time constant values and photointermediate spectra were obtained via global analysis of the data sets. The sequential model explored spectral evolution and produced the EADS, representing the species-associated difference spectra[87].

UV/vis data obtained from FTIR samples were analyzed using custom code implemented in Octave 4.2. and MATLAB R2016b.

Stationary absorption spectra were analyzed using Origin 2017 (OriginLab), normalized to maximum absorption at 280 nm or maximum chromophore absorption, smoothed using Savitzki-Golay method using a 10-point window and 5th order polynomial function. Experimental pk$_a$-values were determined with a Boltzmann function.

FTIR difference spectra were preprocessed using OPUS 7.5 software (Bruker Optics). FTIR data were analyzed via single value decomposition and rotation procedure and subsequent global fit algorithm implemented in Octave 4.2.[88,89]. Assuming a sequential reaction scheme, a sum of exponential functions was used as the fit model.

RR data was background subtracted with custom written software using a polynomial function and further analyzed using the LabSpec Spectroscopy Suite (Horiba).

VMD[90] and PyMol 2.2.3 (Schrödinger, NY) were used to analyze and visualize MD simulation results and computed ion permeation pathways.

Neurophysiological data were analyzed and plotted in Igor Pro 8.0. (wavemetrics, Lake Oswego, OR). Neuronal imaging data was analyzed using ScanImage 2017b (Vidrio Technologies) and Fiji software[91].

If not stated otherwise, data was plotted using either MATLAB R2016b (The MathWorks), GraphPad Prism 7.0 (GraphPad Software Inc., San Diego, CA) or Origin 2017 (OriginLab). Final esthetical adjustments were performed using Adobe Illustrator 2017 (Adobe Systems, San José, CA) or Affinity Designer 1.6 (Serif, Nottingham, UK)

No statistical tests were used to predetermine sample size. Sample sizes were similar to those commonly used in this research field. Repeated experiments always refer to biological replicates performed using at least two batches of transfected cell cultures. Data is given as mean ± standard deviation. Single measurement data, exact sample size ($n$) for each experimental group/condition and further statistical analysis[92] are provided in the Supplementary Figs. 3, 4. Blinding was not performed to ensure correct assignment of the data to the measured constructs and/or experimental conditions. However, randomization was performed in case of buffer exchange experiments and automated analysis was used whenever possible.

**Reporting summary**. Further information on research design is available in the Nature Research Reporting Summary linked to this article.

## Data availability

Data supporting the findings of this manuscript are available from the corresponding authors upon reasonable request. A reporting summary for this Article is available as a Supplementary Information file. The source data underlying Figs. 1d–g, 2b–d,e–g, i, j, 4e–i and Supplementary Figs. S1, 3b, e, 4a–I, 5d,h are provided as a Source Data file.

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

## Acknowledgements

We thank T. Tharmalingam, S. Augustin, M. Reh, M. Meiworm, K. Sauter, and S. Schillemeit for excellent technical assistance. We thank S. P. Tsunoda and H. Kandori for sharing the *GtCCR4* plasmid and Itai Sharon for initial *Tara* Ocean assemblies. This work was supported by the Deutsche Forschungsgemeinschaft (DFG; German Research Foundation): SFB 1078 to P.He. (B1, B2), F.B. (B5), and P.Hi. (B6), the Germany's Excellence Strategy – EXC 2008/1 (UniSysCat) – 390540038 to P.Hi. & P.He., and the priority program SPP 1926 & FOR 2419 to J.S.W. This work was also supported by the European Research Council (ERC): advanced grant (LS1, ERC-2015-AdG) to P.He. and starting grant (LS5, ERC-2016-STG) to J.S.W. P.He. is a Hertie Senior Professor for Neuroscience supported by the Hertie Foundation. O.B. is supported by the Louis and Lyra Richmond Memorial Chair in Life Sciences.

## Author contributions

J.W., J.O., and P.He. designed the project, with contributions from all authors. J.F-U., E. S., O.B., J.W., and J.O. performed computational work. J.O., B.L., S.R-R., P.F., J.W., A.S., A.K., and J.V. conducted the experiments, with assistance from M.B. and M.L. Experimental data were analyzed by J.O., J.W., B.L., S.R-R., P.F., A.S., and A.K. and interpreted by all authors. J.W., J.O., and P.He. wrote the paper, with contributions from all authors.

## Additional information

**Competing interests:** The authors declare no competing interests.

