## [Peer Review File · Nature Communications]

Reviewers' Comments:

Reviewer #1:

Remarks to the Author:

The manuscript by Oppermann et al. entitled "MerMAIDS: A novel family of metagenomically discovered, marine, anion-conducting and intensely desensitizing channelrhodopsins" is a well-written description of a study in which a new class of channel rhodopsins (ChRs) were discovered and characterized spectroscopically and electrophysiologically. The channels were identified metagenomically and form a phylogenetically distinct family of anion selective ChRs most closely related to Chlorophyte cation-conducting ChRs. The most distinguishing unique feature of the channels in this new family is that they almost completely desensitize rapidly after reaching light-evoked maximum current, making them of great interest as optogenetic silencers for transient suppression of individual action potentials without affecting subsequent spikes under continuous exposure to light. Intriguingly, complete desensitization appears to be controlled by a single amino acid residue, the highly conserved Cys84. Changing Cys84 to Thr rescues the more typical desensitization phenotype of roughly 60% inactivated during continuous illumination. The mutation also prolongs dramatically recovery in the dark.

The study is well designed and executed, the results clear cut, and the conclusions rest soundly on the results. The topic is timely, and the paper will be of interest to a broad readership of Nature Communications.

A minor suggestion for the authors to consider: include transient spectral data for the C84T mutant if available.

Reviewer #2:

Remarks to the Author:

The submitted manuscript deserves publication in Nature Communications. In fact, the manuscript reports on the results of an interdisciplinary study comprising the discovery and physiological and spectroscopic characterization of a new clade of microbial marine rhodopsins. The discovered clade includes anion-selective light-gated ion channels which differ from the previously reported channelrhodopsins in terms of deactivation kinetic and efficiency. Most importantly, the authors show that this peculiar new properties can be used in novel optogenetic applications requiring tools capable to effectively switch off a single, impulsively triggered, action potential under continuous illumination.

The authors also report on possible residue-level mechanistic hypotheses for: (i) the anion-selectivity function and (ii) the effective deactivation grounded on both mutational, spectroscopic and computational (modeling) studies. While, these hypotheses represent the less solid part of the manuscript, they provide information sufficient to justify publication provided the authors clarify/respond the points listed below:

Lines 92-94. "...However, sequence comparisons indicated that MerMAIDs might be anion-conducting due to the lack of typical glutamate residues found in chlorophyte CCRs, though also missing in chrytophyte CCRs (Fig. S2)...". I believe that a new sentence, with references, has to be added to explain why the lack of certain glutamate residues would support selective anion transport. The reported sentence, as it is now, will leave the reader puzzled (i.e. the reader is not supposed to know the mechanistic details of channelrhodopsin's gating).

Line 126. "... Cl⁻ efflux ..." should be more clearly "...Cl⁻ outward flux ...". More in general, since the manuscript report on a multidisciplinary research, technical terms and jargon specific of a certain field has to be somehow defined or better explained in the context of the reported results. For instance terms such as "reverse potential upshift" or, later (line 144) , "holding potential

"needs to be clarified as much as possible for readers with a biochemistry or chemistry background.

Line 159. "red-shifted" should be "blue-shifted".

Line 196. "isoforms" -> "isomers" (or, alternative, indicate which exact structural changes the terms "isoforms" indicates ... like conformational or configurational changes).

Lines 207-208. "...mixture of two chromophore isomers, corresponding to the protonated and deprotonated all-trans RSB of the dark state at high pH...". The authors should use correct terms. The protonated and unprotonated forms of the retinal chromophore are NOT isomers (in chemistry). Use for instance "mixture of chromophore cationic and neutral forms".

Line 255. If E44 is originally charged (as the authors seem to assume) then the expected shift upon E44Q mutation must be larger than 3 nm. This experimental fact points to a neutral E44 (in spite of the calculated E44 which must have a large error bar) and a charged D210.

Lines 258-259. Also, the ca. 13 nm red-shift seen when mutating both putative counterions E44 and D210, may reflect the elimination of a single negatively charged residue and its substitution with two different polar residues (Q44 and Q210) - i.e. a charge - Schiff base electrostatic interaction has been replaced with two dipoles - Schiff base interactions).

Lines 269-273. This sentence has to be rephrased/clarified: "...C84 might stabilize the deprotonated RSB within the anion permeation pathway instead, thereby disrupting further ion conduction via charge repulsion of the deprotonated RSB during the photocycle...". The chemistry conveyed by this sentence cannot be understood. C84 has a dipolar side-chain with a polarized S-H bond. It is thus not clear how such a bond can stabilize a deprotonated (i.e. neutral) Schiff base. Which interaction or mechanisms would make such stabilization possible? Do the authors refer instead to a destabilization of the protonated form due to the C84 residue, thus shifting the equilibrium towards a deprotonated chromophore isomer that cannot complete the photocycle?

Lines 324-325. Does the following sentence "...Chloride flux in MerMAID1 is interrupted by charge repulsion following deprotonation of the Schiff base..." mean ".....Chloride flux in MerMAID1 is interrupted by lack of negative charge attraction following deprotonation of the Schiff base..." or, alternatively, ".....Chloride flux in MerMAID1 is interrupted by charge repulsion from D201 following deprotonation of the Schiff base and therefore larger negative charge repulsion?. Please, clarify and/or avoid speculations which may result confusing.

Line 339. Is this due to the change in absorption max? Or to the fact that the 13-cis isomer is not capable to photochemically revert back? Is there a mechanistic idea behind this claim?

Lines 348-350. There is a lack of mechanistic explanation for point i and, also, the authors need to remind the readership about the need of an anti to syn isomerization for closing the photocycle.

Lines 360-362. "...RR spectra of MerMAID1 at pH 10 (Fig. S5e) suggested that the retinal is similarly deprotonated and adopts a rigid configuration in all-trans instead of 13-cis, as in the M-state...". These appear just mechanistic hypothesis that the authors may decide to avoid mentioning.

Line 505. Swiss-model is not the software (methodology) of choice when constructing homology (comparative) models of membrane proteins and rhodopsins (the authors should review the traditional and more modern literature regarding this point). The final model would be therefore a rough model whose quality is difficult to judge.

Reviewer #3:

Remarks to the Author:

The manuscript by Oppermann et al. reports the discovery of a new family of channelrhodopsins, named MerMAIDs, in metagenomic data collected by the Tara oceans project. These proteins conduct anions under illumination, and yet their primary sequences show a considerable difference from those of the earlier known anion-conducting channelrhodopsins from cryptophyte algae. This makes them particularly valuable for research into structural determinants of anion selectivity in channelrhodopsins, which have yet to be identified. The Authors can be congratulated on their thorough, multidisciplinary characterization of MerMAIDs by a combination of electrophysiological, optical and bioinformatics methods, including the demonstration of the utility of these proteins as optogenetic tools for time-resolved silencing of individual neuronal spikes. I recommend this manuscript for publication in Nature Communications after a minor revision. The issues that need to be resolved in the revised version are listed below.

Major issues:

Page 13, line 240: "The D210, E44, W80, and Y48 side chains..." and throughout the text.

This numbering of amino acid residues in MerMAID1 sequence corresponds not to their positions in the primary sequence, but to those in the alignment shown in Figure S2, which contains gaps. Therefore, the actual number of E44 is E41, that of Y48 is Y45 and so on. This is absolutely inappropriate, because different alignments may contain different number of gaps in different positions, and the same residue will turn out having different numbers depending of what sequences it was aligned with. The Authors should correct the numbers according to the primary sequence without gaps.

Page 17, lines 338-339: "This mechanism is consistent with an M-state that cannot be photochemically converted back to the dark state (Fig. S5g,h)."

This is a very strong conclusion. However, it is based entirely on indirect photocurrent measurements. An alternative explanation for the absence of the blue light effect on photocurrent can be that the protonation state of the RSB is not the only factor in channel closing. In any case, the absence of M photoactivity should be confirmed by direct measurements of absorption changes in purified pigment, or this statement should be deleted from the text.

Page 16, line 324-325: "Chloride flux in MerMAID1 is interrupted by charge repulsion following deprotonation of the Schiff base" (and also a similar statement on Page 14, lines 271-273).

It is not clear how the deprotonated (i.e. neutral) Schiff base may cause charge repulsion.

Minor issues:

Page 7, line 132: "Replacement of Cl⁻ with Br⁻ or NO₃⁻ resulted in negative..."

This observation actually shows not only that the channel is non-selective among anions, as the Authors concluded, but that its relative permeability follows the sequence Cl⁻ < Br⁻ < NO₃⁻, as that of the channels reported in Ref. 21 and 23, which is worth noting in the text.

Page 8, line 144: "...whereas τ_{des} was accelerated for MerMAIDs 2, 6..."

The time constant does not accelerate, it increases.

Page 16, line 315-316: "The K→L transition and formation of the open state is accompanied by minimal protein backbone changes..."

According to Fig. 3c, the half-decay time of K is below 2×10^{-7} s, whereas the half-rise time of photocurrent is 10^{-4} s. It is not clear how the Authors explain this difference. Also, the Authors should explain the reasons for their conclusion that backbone changes are minimal.

Table S2: The Authors do not provide GenBank accession numbers for the MerMAID sequences. It is understandable that they do not want the sequences to be released to the public before publication of their paper, but the readers would need these numbers after the paper is published. The Authors should provide the numbers and ask GenBank not to release the sequences to the public until the paper is accepted for publication. The number of CrChR2 is also missing.

Reviewer #1 (Remarks to the Author):

The manuscript by Oppermann et al. entitled “MerMAIDS: A novel family of metagenomically discovered, marine, anion-conducting and intensely desensitizing channelrhodopsins” is a well-written description of a study in which a new class of channel rhodopsins (ChRs) were discovered and characterized spectroscopically and electrophysiologically. The channels were identified metagenomically and form a phylogenetically distinct family of anion selective ChRs most closely related to Chlorophyte cation-conducting ChRs. The most distinguishing unique feature of the channels in this new family is that they almost completely desensitize rapidly after reaching light-evoked maximum current, making them of great interest as optogenetic silencers for transient suppression of individual action potentials without affecting subsequent spikes under continuous exposure to light. Intriguingly, complete desensitization appears to be controlled by a single amino acid residue, the highly conserved Cys84. Changing Cys84 to Thr rescues the more typical desensitization phenotype of roughly 60% inactivated during continuous illumination. The mutation also prolongs dramatically recovery in the dark.

The study is well designed and executed, the results clear cut, and the conclusions rest soundly on the results. The topic is timely, and the paper will be of interest to a broad readership of Nature Communications.

We thank reviewer #1 for the encouraging comments on our manuscript. Our detailed answers can be found below.

A minor suggestion for the authors to consider: include transient spectral data for the C84T mutant if available.

We agree that transient spectral data of the C84T mutant would be highly interesting. Unfortunately, we were not able to produce purified protein of the C84T mutant and thus cannot provide requested data. We now included a statement that the requested data would be of interest for future studies.

Reviewer #2 (Remarks to the Author):

The submitted manuscript deserves publication in Nature Communications. In fact, the manuscript reports on the results of an interdisciplinary study comprising the discovery and physiological and spectroscopic characterization of a new clade of microbial marine rhodopsins. The discovered clade includes anion-selective light-gated ion channels which differ from the previously reported channelrhodopsins in terms of deactivation kinetic and efficiency. Most importantly, the authors show that this peculiar new properties can be used in novel ontogenetic applications requiring tools capable to effectively switch off a sigle, impulsively triggered, action potential under continuous illumination.

The authors also report on possible residue-level mechanistic hypotheses for: (i) the anion-selectivity function and (ii) the effective deactivation grounded on both mutational, spectroscopic and computational (modeling) studies. While, these hypotheses represent the less solid part of

the manuscript, they provide information sufficient to justify publication provided the authors clarify/respond the points listed below:

We are glad to have received a positive assessment from reviewer #2 and are grateful for constructive criticism that will be answered in detail below.

Lines 92-94. "...However, sequence comparisons indicated that MerMAIDs might be anion-conducting due to the lack of typical glutamate residues found in chlorophyte CCRs, though also missing in chrytophyte CCRs (Fig. S2)...". I believe that a new sentence, with references, has to be added to explain why the lack of certain glutamate residues would support selective anion transport. The reported sentence, as it is now, will leave the reader puzzled (i.e. the reader is not supposed to know the mechanistic details of channelrhodopsin's gating).

We agree that this section was indeed not ideally written. We revised this section and provide requested references.

Now from line 94:

"... Sequence comparisons, however, indicated that MerMAIDs might be anion-conducting due to the lack of typical glutamate residues found in chlorophyte CCRs (Fig. S2). As already shown, replacement of pore-lining glutamates with positively charged or neutral amino acids can mediate anion selectivity in originally cation-conducting chlorophyte CCRs¹⁸⁻²². Nevertheless, cryptophyte CCRs were shown to conduct cations although lacking typical glutamate motives by operating with an alternative mode more related to light-driven rhodopsin ion pumps³³⁻³⁵. To examine the MerMAIDs function and ion selectivity, we expressed them in human embryonic kidney (HEK) cells and performed whole-cell voltage-clamp experiments at 1-day post transfection."

Line 126. "... Cl⁻ efflux ..." should be more clearly "...Cl⁻ outward flux ...". More in general, since the manuscript report on a multidisciplinary research, technical terms and jargon specific of a certain field has to be somehow defined or better explained in the context of the reported results. For instance terms such as "reverse potential upshift" or, later (line 144) , "holding potential "needs to be clarified as much as possible for readers with a biochemistry or chemistry background.

For simplified understanding we replaced the mentioned terms within our revised manuscript and provide a better definition of terms as mentioned below.

Now from line 130:

"Next, we tested the ion selectivity of the MerMAIDs. Therefore, photocurrents at different membrane potentials where recorded to deduce the reversal potential (E_{rev} ; the potential where the net ion flux is zero) in different ionic conditions (Fig. 2a,b). Because we suspected anion selectivity, we depleted the extracellular Cl⁻ from 150 mM to 10 mM while maintaining the intracellular Cl⁻ at 120 mM (Fig. 2a). This increased the inward current and induced a positive shift of the reversal potential (ΔE_{rev} , Fig. 2a,b), consistent with a Cl⁻ outward flux. A similar shift close to the theoretical Cl⁻-Nernst potential was obtained for all MerMAIDs (Figs. 2c and S4a), as well as for the small stationary photocurrents of MerMAID1 (Figs. 2a,c and S4a). These data justified the classification of MerMAIDs as ACRs. "

Moreover, we replaced "holding potential" by "membrane potential" throughout the manuscript, to avoid confusion of the reader.

Line 159. "red-shifted" should be "blue-shifted".

Corrected.

Line 196. "isoforms" -> "isomers" (or, alternative, indicate which exact structural changes the terms "isoforms" indicates ... like conformational or configurational changes).

Corrected.

Lines 207-208. "...mixture of two chromophore isomers, corresponding to the protonated and deprotonated all-trans RSB of the dark state at high pH...". The authors should use correct terms. The protonated and unprotonated forms of the retinal chromophore are NOT isomers (in chemistry). Use for instance "mixture of chromophore cationic and neutral forms".

We removed the term "isoform" accordingly.

Now from line 215:

"Notably, strong bands in the C=C stretching region at ca. 1540 and 1577 cm^{-1} indicated a mixture of the protonated and deprotonated all-trans RSB of the dark state at high pH."

Line 255. If E44 is originally charged (as the authors seem to assume) then the expected shift upon E44Q mutation must be larger than 3 nm. This experimental fact points to a neutral E44 (in spite of the calculated E44 which must have a large error bar) and a charged D210.

Please see comment below.

Lines 258-259. Also, the ca. 13 nm red-shift seen when mutating both putative counterions E44 and D210, may reflect the elimination of a single negatively charged residue and its substitution with two different polar residues (Q44 and Q210) - i.e. a charge - Schiff base electrostatic interaction has been replaced with two dipoles - Schiff base interactions).

As correctly pointed out by Reviewer #2, the slight bathochromic shift is indicative for a protonated E44 in the wild type. In the initially submitted manuscript version we only stated that the pK calculation predicted a deprotonated E44. We now included the following for clarification:

Now from line 262:

"...Elimination of the more distant E44 via an E44Q mutation caused only a 3 nm bathochromic action-spectrum shift (Fig. 4f), indicative for a protonated E44 in wild-type MerMAID1 unlike predicted from our model structure pK_a calculations. Moreover, the E44Q mutation increased the photocurrent amplitudes (Fig. 4f),..."

Lines 269-273. This sentence has to be rephrased/clarified: "...C84 might stabilize the deprotonated RSB within the anion permeation pathway instead, thereby disrupting further ion conduction via charge repulsion of the deprotonated RSB during the photocycle...". The chemistry conveyed by this sentence cannot be understood. C84 has a dipolar side-chain with a polarized S-H bond. It is thus not clear how such a bond can stabilize a deprotonated (i.e. neutral) Schiff base. Which interaction or mechanisms would make such stabilization possible?

Do the authors refer instead to a destabilization of the protonated form due to the C84 residue, thus shifting the equilibrium towards a deprotonated chromophore isomer that cannot complete the photocycle?

We thank Reviewer #2 for pointing to this. We deleted the particular sentence from this paragraph and revised the discussion regarding the role of C84:

Now from line 355:

“We found that replacing C84 in MerMAID1 decreases the peak photocurrent but increases stationary photocurrents, thus reducing the extent of desensitization, possibly due to either a prolonged L-state or a shortened recovery from the M- to the initial dark state. However, in the C84T mutant, the desensitization kinetic was found to be accelerated and recovery to the dark state slowed down. Hence, stationary photocurrents might be potentially a cause of a branched photocycle instead, in which a second photoactive closed state can be populated from the initial dark state by photon absorption, as discussed for various CCRs^{7,52,8,53,54}. As recently substantiated experimentally for CrChR2, the retinal in both parallel photocycles differ, adopting either a syn or an anti conformation of the retinal chromophore¹⁰. Conceivably, C84 could suppress a C=N anti to syn isomerization in MerMAID1 and therefore prevent population of parallel syn-photocycle that could account for a second conducting state and stationary photocurrents. However, future studies are necessary to prove presence of various retinal isomers during the photocycle of MerMAID1 C84T.”

As also requested by reviewer #1, it would be indeed interesting to get spectral data of the C84T mutant. However, we were unable to purify this mutant, unfortunately.

Lines 324-325. Does the following sentence "...Chloride flux in MerMAID1 is interrupted by charge repulsion following deprotonation of the Schiff base..." means ".....Chloride flux in MerMAID1 is interrupted by lack of negative charge attraction following deprotonation of the Schiff base..." or, alternatively, ".....Chloride flux in MerMAID1 is interrupted by charge repulsion from D201 following deprotonation of the Schiff base and therefore larger negative charge repulsion?. Please, clarify and/or avoid speculations which may result confusing.

As this is part of the discussion, potential mechanisms should be discussed. However, we agree that this particular part could lead to confusion about the mechanism and thus, we revised it accordingly as suggested:

Now from line 332:

“Chloride flux in MerMAID1 is interrupted by lack of negative charge attraction following deprotonation of the Schiff base.”

Line 339. Is this due to the change in absorption max? Or to the fact that the 13-cis isomer is not capable to photochemically revert back? Is there a mechanistic idea behind this claim?

Both suggestions can contribute to the mechanism. If the non-conducting state could be depopulated (light induced), accompanied by repopulation of the dark state, a fraction of MerMAID molecules would

be available to re-enter the photocycle and cause stationary photocurrents. As pointed out, the strong blueshift of desensitized state could account for this as it is simply not absorbing at wavelengths around 500 nm. However, we showed that also application of UV light could not cause any change in photocurrents or recovery kinetics (Fig. S5g,h). As requested by Reviewer #3, we now also included IR spectroscopic data, directly showing that indeed the 13-*cis* isomer is not capable of reverting back to the ground state when illuminated with UV light (Fig S5i). Anyhow, it is not clear why a photon induced back reaction is not possible in MerMAID1. We revised our discussion accordingly.

Now from line 346:

“...This mechanism is consistent with an M-state that cannot be photochemically converted back to the dark state, which is the case for MerMAID1 as demonstrated by electrophysiological and IR spectroscopic data (Fig. S5g-i), while the underlying mechanism is not understood....”

Lines 348-350. There is a lack of mechanistic explanation for point i and, also, the authors need to remind the readership about the need of an anti syn isomerization for closing the photocycle.

We have revised this section as shown below.

Now from line 355:

“We found that replacing C84 in MerMAID1 decreases the peak photocurrent but increases stationary photocurrents, thus reducing the extent of desensitization, possibly due to either a prolonged L-state or a shortened recovery from the M- to the initial dark state. However, in the C84T mutant, the desensitization kinetic was found to be accelerated and recovery to the dark state was slowed down. Hence, stationary photocurrents might be potentially a cause of a branched photocycle instead, in which a second photoactive closed state can be populated from the initial dark state by photon absorption, as discussed for various CCRs^{7,52,8,53,54}. As recently substantiated experimentally for CrChR2, the retinal in both parallel photocycles differ, adopting either a syn or an anti conformation¹⁰. Conceivably, C84 could suppress a C=N anti to syn isomerization in MerMAID1 and therefore prevent population of parallel syn-photocycle that could account for a second conducting state and stationary photocurrents. However, future studies are necessary to prove presence of various retinal isomers during the photocycle of MerMAID1 C84T.”

In addition, we now remind the reader to the necessary *cis-trans* reisomerization of the chromophore during the photocycle in another section of the discussion.

Now from line 333:

“In the final photocycle step, MerMAID1 structurally rearranges, the RSB reprotonates and reisomerizes to all-trans during recovery of the initial dark state that is fully repopulated within seconds.”

Lines 360-362. “...RR spectra of MerMAID1 at pH 10 (Fig. S5e) suggested that the retinal is similarly deprotonated and adopts a rigid configuration in all-trans instead of 13-cis, as in the M-state....”. These appear just mechanistic hypothesis that the authors may decide to avoid mentioning.

We agree that this is statement is a hypothesis and revised this sentence towards clarifying that our conclusion is hypothetical.

Now from line 377:

“RR spectra of MerMAID1 at pH 10 (Fig. S5e) possibly suggested that the retinal is similarly deprotonated and adopts a rigid configuration in all-trans instead of 13-cis, as in the M-state.”

Line 505. Swiss-model is not the software (methodology) of choice when constructing homology (comparative) models of membrane proteins and rhodopsins (the authors should review the traditional and more modern literature regarding this point). The final model would be therefore a rough model whose quality is difficult to judge.

We agree that SwissModel is not state of the art software for modeling membrane proteins. Therefore, we subsequently performed a MD simulation of the model embedded in a phospholipid bilayer. As shown by our 101 ns simulation, the side chains of the active-site residues re-orient at the beginning of our simulations and are comparatively stable over the following time course of the MD simulation, which verifies that the MD simulated model (not the initial Swiss-model) rationally describes the active site complex.

To further test our model, we compared it with other software for modelling membrane proteins, namely Memoir (Ebejer et al. 2013), MEDELLER (Kelm et al. 2010) and the Robetta server (Song et al. 2013, Raman et al. 2009, Kim et al. 2004) and compared the active-site residues of the initial and the MD simulated model with the results of the above-mentioned modeling tools. Notably, we observed only small deviations between the models. Below root mean square deviation (RMSD) values (in Å) of the active-site residues between the different models can be found:

	SwissModel	MD
SwissModel	-	0.46
MD	0.46	-
Memoir	0.12	0.49
MEDELLER	0.30	0.52
Robetta	0.55-0.76	0.63-0.92

Please note that the Robetta server predicts 5 models in total. Best and worst model RMSD values are shown accordingly in the above table. For your convenience we also show the best matching Robetta

model (orange), compared to the initial (not MD simulated) SwissModell (gray).

As can be deduced from the comparison above, the side chain conformation between the models is comparable, although the Robetta model performs worse than the Memoir/MEDELLER models. More importantly, the new models do not contain the Schiff base linked retinal and water molecules that we already provided with our MD simulation. As the differences between the models are small, we surmise that further MD simulations on the new models would end up with a very similar result regarding the previously presented model structure. We will therefore adhere to the model we already present.

Reviewer #3 (Remarks to the Author):

The manuscript by Oppermann et al. reports the discovery of a new family of channelrhodopsins, named MerMAIDs, in metagenomic data collected by the Tara oceans project. These proteins conduct anions under illumination, and yet their primary sequences show a considerable difference from those of the earlier known anion-conducting channelrhodopsins from cryptophyte algae. This makes them particularly valuable for research into structural determinants of anion selectivity in channelrhodopsins, which have yet to be identified. The Authors can be congratulated on their thorough, multidisciplinary characterization of MerMAIDs by a combination of electrophysiological, optical and bioinformatics methods, including the demonstration of the utility of these proteins as optogenetic tools for time-resolved silencing of individual neuronal spikes. I recommend this manuscript for publication in Nature Communications after a minor revision. The issues that need to be resolved in the revised version are listed below.

We thank Reviewer #3 for the congratulations and comments regarding our manuscript. Our detailed point-to-point response can be found below.

Major issues:

Page 13, line 240: “The D210, E44, W80, and Y48 side chains...” and throughout the text.

This numbering of amino acid residues in MerMAID1 sequence corresponds not to their

positions in the primary sequence, but to those in the alignment shown in Figure S2, which contains gaps. Therefore, the actual number of E44 is E41, that of Y48 is Y45 and so on. This is absolutely inappropriate, because different alignments may contain different number of gaps in different positions, and the same residue will turn out having different numbers depending of what sequences it was aligned with. The Authors should correct the numbers according to the primary sequence without gaps.

We apologize for any confusion caused by the residue numbering. The numbering in Figure S2 is actually correct. Gaps are not counted in this numbering. The alignment is cropped N- and C-terminally. Hence, numbering appeared to be wrong. We updated the legend of Figure S2 to avoid confusion of the reader. Moreover, we now provide numbering for all ChRs on the right hand side of Figure S2 for better comparison between different ChRs.

Page 17, lines 338-339: “This mechanism is consistent with an M-state that cannot be photochemically converted back to the dark state (Fig. S5g,h).”

This is a very strong conclusion. However, it is based entirely on indirect photocurrent measurements. An alternative explanation for the absence of the blue light effect on photocurrent can be that the protonation state of the RSB is not the only factor in channel closing. In any case, the absence of M photoactivity should be confirmed by direct measurements of absorption changes in purified pigment, or this statement should be deleted from the text.

We agree that our electrophysiological measurements were not a direct readout. We now included new IR spectroscopic data where we applied UV-light in addition to continuous green-light illumination of MerMAID1 (Fig. S5i). We could show that 362 nm illumination has indeed no effect, neither on the fast, nor the slow kinetic IR component, indicating that the M-state is not photoreactive. We agree that the protonation state of the RSB might not be the sole determinant of channel closing and revised our discussion accordingly.

Page 16, line 324-325: “Chloride flux in MerMAID1 is interrupted by charge repulsion following deprotonation of the Schiff base” (and also a similar statement on Page 14, lines 271-273).

It is not clear how the deprotonated (i.e. neutral) Schiff base may cause charge repulsion.

As already pointed out in our response to reviewer #2, we deleted the statement on page 14 and revised the discussion accordingly:

Now from line 332

“Chloride flux in MerMAID1 is interrupted by lack of negative charge attraction following deprotonation of the Schiff base”

Minor issues:

Page 7, line 132: “Replacement of Cl⁻ with Br⁻ or NO₃⁻ resulted in negative...”

This observation actually shows not only that the channel is non-selective among anions, as the

Authors concluded, but that its relative permeability follows the sequence $Cl^- < Br^- < NO_3^-$, as that of the channels reported in Ref. 21 and 23, which is worth noting in the text.

We totally agree that the permeability statement should be included. We updated the paragraph accordingly

Now from line 141:

“...thus revealing nonselective anion conductivity with a relative permeability sequence that follows $Cl^- < Br^- < NO_3^-$, as previously reported...”

Page 8, line 144: “...whereas τ_{des} was accelerated for MerMAIDs 2, 6...”

The time constant does not accelerate, it increases.

As correctly mentioned, τ_{des} cannot accelerate but if the desensitization becomes faster, the time constant decreases. We modified the text accordingly.

Now from line 153:

“...whereas τ_{des} decreased for MerMAIDs 2, 6, and 7 (Fig. 2g).”

Page 16, line 315-316: “The K→L transition and formation of the open state is accompanied by minimal protein backbone changes...”

According to Fig. 3c, the half-decay time of K is below 2×10^{-7} s, whereas the half-rise time of photocurrent is 10^{-4} s. It is not clear how the Authors explain this difference. Also, the Authors should explain the reasons for their conclusion that backbone changes are minimal.

The transition from K to L is indeed much faster than the rise of the photocurrent. To avoid an interpretation where K→L is correlated with channel opening, we revised this part of the discussion accordingly and also modified our interpretation regarding the minimal backbone changes (including the supplementary discussion).

Now from line 322:

“..., analogous to bacteriorhodopsin³⁷. Following the K→L transition, the L state remains UV/vis spectroscopically unchanged over almost three temporal orders of magnitude, while channel opening proceeds during the long-lived L-state. Accordingly, and in line with FTIR data (see Supplemental Discussion), only minimal protein backbone changes involving residues in the vicinity of the RSB can take place during formation of the conducting state. Maximum...”

Table S2: The Authors do not provide GenBank accession numbers for the MerMAID sequences. It is understandable that they do not want the sequences to be released to the public before publication of their paper, but the readers would need these numbers after the paper is published. The Authors should provide the numbers and ask GenBank not to release the sequences to the public until the paper is accepted for publication. The number of CrChR2 is also missing.

We fully agree that GenBank entries are necessary. We have generated GenBank entries and updated table S2 and the data availability statement accordingly. Further we added the missing information for *CrChr2*. Please note that it takes some time until the GenBank IDs are accessible to the public.

Reviewers' Comments:

Reviewer #2:

Remarks to the Author:

I have no further comments/requests. I recommend the revised manuscript for publication without further changes.

Reviewer #3:

Remarks to the Author:

I am satisfied with the Authors' response and recommend the revised manuscript for publication.